# Anti-PD-1 Monoclonal Antibodies (mAbs) Are Superior to Anti-PD-L1 mAbs When Combined with Chemotherapy in First-Line Treatment for Metastatic Non-Small Cell Lung Cancer (mNSCLC): A Network Meta-Analysis

**DOI:** 10.3390/biomedicines11071827

**Published:** 2023-06-26

**Authors:** Joe Q. Wei, Alexander Yuile, Malinda Itchins, Benjamin Y. Kong, Bob T. Li, Nick Pavlakis, David L. Chan, Stephen J. Clarke

**Affiliations:** 1Royal North Shore Hospital, St Leonards, NSW 2065, Australia; alexander.yuile@health.nsw.gov.au (A.Y.); malinda.itchins@sydney.edu.au (M.I.); ben.kong@sydney.edu.au (B.Y.K.); nick.pavlakis@sydney.edu.au (N.P.); david.chan@sydney.edu.au (D.L.C.); stephen.clarke@sydney.edu.au (S.J.C.); 2Northern Clinical School, University of Sydney, St Leonards, NSW 2065, Australia; 3Memorial Sloan Kettering Cancer Centre, Weill Cornell Medical College, New York, NY 10065, USA; lib1@mskcc.org

**Keywords:** immunotherapy, chemoimmunotherapy, non-small cell lung cancer, anti-PD-1, anti-PD-L1

## Abstract

Platinum-based chemotherapy combined with anti-PD-1 or PD-L1 monoclonal antibodies (mAbs) is now standard first-line therapy for mNSCLC patients without sensitizing driver mutations. Anti-PD-1 and anti-PD-L1 mAbs are considered to be equivalent in efficacy. In the absence of head-to-head randomized control trials (RCTs), we utilized network meta-analysis (NWM) to provide an indirect comparison of their efficacy. A systematic literature review and NWM were performed using RCTs that investigated anti-PD-1 or PD-L1 mAbs ± chemotherapy in patients with mNSCLC in the first-line setting. The primary outcome was comparative overall survival (OS), while secondary outcomes were comparative progression-free survival (PFS), objective response rate (ORR), and rate of grade 3 and higher toxicities. We identified 24 RCTs. Patients treated with anti-PD-1 mAb + chemotherapy compared with anti-PD-L1 mAb + chemotherapy showed superior mOS, mPFS, and ORR with a similar rate of grade 3 and higher toxicities. This difference in mOS was most pronounced in the PD-L1 TPS 1–49% population. The two mAbs were equivalent as single agents. Anti-PD-1 mAb + chemotherapy improved mOS when compared to anti-PD-1 mAb monotherapy, whereas anti-PD-L1 mAbs + chemotherapy did not when compared to anti-PD-L1 mAb monotherapy. Head-to-head RCTs are warranted in the future.

## 1. Introduction

Lung cancer is the leading cause of cancer death worldwide [1]. Approximately 85% of occurrences are non-small cell lung cancers (NSCLC), of which 70% are diagnosed at an advanced or metastatic stage [2], defining a need for effective systemic therapy. NSCLC can be classified as squamous or non-squamous cell carcinoma. Platinum-based doublet chemotherapy has been the backbone treatment for metastatic NSCLC (mNSCLC) for decades.

Immune checkpoint inhibitors (ICIs) have revolutionized cancer therapies. Monoclonal antibodies (mAbs) that inhibit programmed death protein 1 (PD-1) or its ligand (PD-L1) are now widely used in various cancers [3,4,5,6]. These mAbs work because they can block the interaction between PD-1 on activated T cells and PD-L1 on tumor cells, thereby reprogramming and reactivating otherwise exhausted anti-tumor T cells [7]. Multiple anti-PD-1 and anti-PD-L1 mAbs have been developed to take advantage of this mechanism. Several large, randomized control trials (RCTs) have shown that to addition of anti-PD-1 or anti-PD-L1 mAb to standard chemotherapy can significantly improve the survival of patients with mNSCLC without sensitizing driver mutations compared with platinum-doublet chemotherapy, and these combinations are now a standard of care [8,9,10].

Because anti-PD-1 and anti-PD-L1 mAbs can both block the PD-1-PD-L1 interaction, their efficacy is thought to be comparable. However, numerous factors could cause differences in efficacy between the two mAbs, including: the greater ability of anti-PD-L1 mAbs (IgG1) to induce antibody-dependent complement-mediated cytotoxicity [11]; the presence of a second ligand for PD-1, PD-L2, which cannot be blocked by anti-PD-L1 mAb [12]; and the potential for chemotherapy to modulate PD-L1 expression within the tumor microenvironment (TME) [13,14]. To date, no head-to-head RCTs comparing PD-1 inhibitor therapy versus PD-L1 as monotherapy or in combination with chemotherapy have been performed to define differences in efficacy between these two classes of mAbs.

In 2021, several network meta-analyses (NWM)—a statistical method used to compare interventions indirectly [15]—have shown that pembrolizumab, an anti-PD-1 mAb, is more effective than atezolizumab, an anti-PD-L1 mAb, when combined with chemotherapy [16,17,18]. Because of the limited number of RCTs available at that time, the authors were unable to conclude if the differences represented an individual drug effect or a drug class effect.

More recently, multiple noval anti-PD-1 and anti-PD-L1 mAbs have been investigated in phase II/III RCTs in the first-line mNSCLC setting [19,20,21,22,23,24,25,26,27,28,29]. In light of these additional studies, we have performed NWM between anti-PD-1 and anti-PD-L1 mAbs with or without platinum-based chemotherapy doublets in an attempt to detect any differences in efficacy between these two mAbs as drug classes.

## 2. Materials and Methods

### 2.1. The Search Strategy and Selection Criteria

This systematic review and NWM were performed according to the Preferred Reporting Items for Systematic Review and Meta-Analyses (PRISMA) Statement. The inclusion criteria and analytical plan were prespecified in a protocol submitted to PROSPERO (registration number CRD42022361215). Eligible studies were retrieved from MEDLINE, PubMed, and EMBASE published up to 1st December 2022. Two authors (JW and AY) developed the search strategy and performed manual searches independently. The following search terms were used to identify studies involving humans: “randomized” AND (“non-small cell lung cancer” OR “lung adenocarcinoma” OR “lung squamous cell carcinoma” OR “lung cancer”) AND (“immunotherapy” OR “PD-L1” OR “PD-1”).

To determine eligibility, titles and abstracts were manually screened by JW and AY independently. The full texts of studies were assessed if eligibility could not be determined by the titles or abstracts. Disagreements regarding eligibility were discussed verbally to reach a consensus. If consensus was not reached, a third author (SC) was involved for the tie-break. Inclusion criteria: phase II or III RCTs involving unresectable, stage IV NSCLC without sensitizing EGFR mutations or ALK rearrangements; an intervention arm involving anti-PD-1 or anti-PD-L1 monoclonal antibodies (mAb) with or without platinum-based chemotherapy; a control arm involving platinum-based chemotherapy; and studies with data on overall survival (OS), progression-free survival (PFS), objective response rate (ORR), and rate of grade 3 and higher toxicities. Exclusion criteria: studies not performed in the first-line setting; studies involving only large cell lung carcinoma or bronchopulmonary neuroendocrine tumors; studies with tyrosine kinase inhibitors, anti-vascular endothelial growth factor (VEGF) inhibitors, or immunotherapies other than anti-PD-1 or anti-PD-L1 mAbs.

### 2.2. Data Collection and Risk of Bias within Individual Studies

The data were extracted by JW and AY separately in accordance with the PRISMA statement. Discrepancies were verbally resolved with SC. Patient characteristics extracted from the eligible studies included sample size, histologic type, and PD-L1 tumor proportion score (TPS). Clinical outcomes extracted included hazard ratios (HR) with 95% confidence intervals (95% CIs) for median OS (mOS) and median PFS (mPFS), the incidence over the total number evaluated (n/N) for ORR, and the rate of grade 3 and higher toxicities.

The risk of bias was assessed using the Cochrane Risk of Bias Tool 2.0 (RoB 2.0) for RCTs [30]. Five domains were assessed for bias: risk of bias arising from the randomization process (D1), risk of bias owing to deviations from the intended interventions (D2), risk of bias from missing outcome data (D3), risk of bias in the measurement of the outcome (D4), and risk of bias in the selection of the reported result (D5). A judgment of risk for each domain was assigned by the selection of one of the three levels of risk of bias for each domain—low risk of bias, some concerns, or high risk of bias. An overall bias assessment was given to the study by considering the results of the above assessments.

### 2.3. Statistical Analysis

The primary outcome was OS; secondary outcomes were PFS and ORR. The effective sizes were HRs with 95% CI for mOS and mPFS, and ORs with 95% CI for ORR and a rate of grade 3 and higher toxicities. Data were collected from studies and entered into REVMAN version 5.4. Using a random effects model, pairwise meta-analyses were performed for the following groups: group 1: anti-PD-1 mAbs + chemotherapy versus chemotherapy; group 2: anti-PD-L1 mAbs + chemotherapy versus chemotherapy; group 3: anti-PD-1 mAbs versus chemotherapy; group 4: anti-PD-L1 mAbs versus chemotherapy (see Figure 1). Subgroup meta-analyses for squamous and non-squamous histology types and PD-L1 TPS < 1%, 1–49%, and >50% were also performed. Studies without data on these subgroups were excluded from subgroup meta-analyses. Heterogeneity was evaluated using the Q test and I^2^ previously described [31]. A Q test *p*-value of >0.1 or I^2^ < 50% was considered to have a low-to-moderate level of heterogeneity; a Q test *p*-value of <0.05 or I^2^ > 50% was considered to have a substantial level of heterogeneity. A sensitivity analysis was performed to investigate the impact of excluding the likely study causing heterogeneity and to see whether the reported result in the pooled analysis was changed. Indirect comparisons of the outcomes described above and subgroup analyses according to PD-L1 TPS were performed between group 1 versus group 2, group 3 versus group 4, group 1 versus group 3, and group 2 versus group 4 (see Figure 1) using the Bayesian NWM described previously [15]. Group 1 versus group 4 and group 2 versus group 3 were not performed as the results could not be interpreted. A *p*-value of <0.05 was considered statistically significant with NMW.

## 3. Results

### 3.1. This Systematic Review and Study Characteristics

We identified 1686 records in the databases. After removing duplicates and ineligible studies by title and abstract screening, 78 full-text articles were retrieved and assessed for eligibility. Finally, 24 RCTs [8,21,22,23,24,26,27,28,29,32,33,34,35,36,37,38,39,40,41,42,43,44,45,46] involving 14,123 patients met the eligibility criteria, and there were no studies with a high risk of bias (see Figure 2 and Figure A1). Sixteen RCTs used anti-PD-1 mAbs and eight used anti-PD-L1 mAbs. Seven different anti-PD-1 mAbs (nivolumab, pembrolizumab, cemiplimab, sintilimab, toripalimab, tislelizumab, and camrelizumab) and four different anti-PD-L1 mAbs (atezolizumab, durvalumab, avelumab, and sugemalimab) were investigated in the first-line setting. We did not identify RCTs that compared anti-PD-1 with anti-PD-L1 mAbs directly.

All 24 RCTs used 4–6 cycles of platinum-based doublet chemotherapy as the control arm. Carboplatin or cisplatin were used as platinum agents in all studies; pemetrexed was used in patients with non-squamous histopathology; and paclitaxel was used in patients with squamous histopathology in all studies. We excluded the following in our analyses to reduce the chance of heterogeneity: an arm involving ipilimumab in CheckMate 227; an arm involving tremelimumab in MYSTIC, and data involving EGFR sensitizing mutations or ALK rearrangements in IMpower 130 (see Table 1).

We identified some differences between the included studies (Table 1). Anti-PD-L1 mAbs used in immunohistochemistry staining (IHC) to classify TPS were different in various studies; IMpower 131 and 132 had a different way of classifying PD-L1 TPS (PD-L1 moderate population was defined as >1% and <50% tumor cell or >1% and <10% immune cell PD-L1 staining, and PD-L1 high population was defined as >50% tumor cell or >10% immune cell PD-L1 staining); only the PD-L1 TPS <1% population in CheckMate 227 received anti-PD-1 mAb + chemotherapy combination (see Table 1). These differences could contribute to heterogeneity in the subgroup analyses (see below).

### 3.2. Pairwise Meta-Analyses for Survival and Tumor Response

#### 3.2.1. Overall Survival

The addition of anti-PD-1 mAb to chemotherapy in patients with mNSCLC improved mOS significantly in patients with mNSCLC compared with chemotherapy alone (HR 0.67, 95% CI 0.62–0.73, *p* < 0.00001; see Figure 3A). Similarly, the addition of anti-PD-L1 mAb to chemotherapy also improved mOS significantly compared with chemotherapy alone (HR 0.83, 95% CI 0.76–0.91, *p* < 0.0001; see Figure 3B). As monotherapies, both anti-PD-1 or anti-PD-L1 mAbs achieved better mOS than chemotherapy in patients with tumors positive for PD-L1 (TPS >1%, see Figure A2; and TPS > 50%, see Figure 3C,D). The results are summarized in Table 2.

#### 3.2.2. Progression-Free Survival

Consistent with mOS, adding anti-PD-1 or anti-PD-L1 mAb to chemotherapy also improved mPFS significantly in patients with mNSCLC compared with chemotherapy alone (anti-PD-1: HR 0.52, 95% CI 0.48–0.57, *p* < 0.00001; anti-PD-L1: HR 0.63, 95% CI 0.55–0.73, *p* < 0.00001; see Figure 4A,B; NB, please see below regarding addressing high levels of heterogeneity). As monotherapies, anti-PD-L1 mAbs were superior to chemotherapy alone (HR 0.76, 95% CI 0.66–0.87, *p* < 0.0001; see Figure A3B), but anti-PD-1 mAbs were not (HR 0.82, 95% CI 0.62–1.09, *p* < 0.18; see Figure A3A). In patients with tumors expressing a high level of PD-L1 (TPS > 50%), both anti-PD-1 and anti-PD-L1 mAb monotherapy improved mPFS when compared with chemotherapy (Figure A3C,D).

#### 3.2.3. Objective Tumor Radiological Response Rate and Grade 3 and Higher Toxcities

Pairwise meta-analyses were also performed for ORR and grade 3 and higher toxicities between anti-PD-1 or anti-PD-L1 mAb ± chemotherapy compared with chemotherapy alone. Both anti-PD-1 and anti-PD-L1 mAbs achieved better ORR when combined with chemotherapy (anti-PD-1: OR 2.54, 95% CI 2.23–2.89, *p* = 0.00001; anti-PD-L1: OR 1.96, 95% CI 1.61–2.39, *p* = 0.00001; see Figure 4C,D), but were equivalent to chemotherapy when used alone (anti-PD-1: OR 1.27, 95% CI 0.81–1.97, *p* = 0.29; anti-PD-L1: OR 1.00, 95% CI 0.82–1.21, *p* = 0.96; see Figure A4). The grade 3 and higher toxicities were similar between ant-PD-1 and anti-PD-L1 mAbs (Figure 4E,F).

### 3.3. Pairwise Meta-Analyses for Overall Survival According to Histology Type and PD-L1 TPS

We predicted that histological subtype and PD-L1 TPS would be sources of possible interstudy heterogeneity; hence, we performed pairwise meta-analyses for these subgroups, and the results are summarized in Table 2. Anti-PD-1 mAb combined with chemotherapy achieved superior mOS regardless of the histological subtype (squamous: HR 0.69, 95% CI 0.59–0.83, *p* = 0.0001; non-squamous: HR 0.67, 95% CI 0.58–0.76, *p* = 0.00001; see Figure A5A,C) and PD-L1 TPS (PD-L1 TPS ≤ 1%: HR 0.76, 95% CI 0.66–0.87, *p* = 0.0001; PD-L1 TPS 1–49%: HR 0.61, 95% CI 0.52–0.71, *p* = 0.00001; PD-L1 TPS > 50%: HR 0.66, 95% CI 0.54–0.81, *p* = 0.0001; see Figure A6A,C,E). In contrast, anti-PD-L1 mAbs combined with chemotherapy achieved superior mOS for patients with non-squamous mNSCLC (HR 0.83, 95% CI 0.73–0.93, *p* = 0.002; see Figure A5D), PD-L1 TPS < 1% (HR 0.85, 95% CI 0.74–0.99, *p* = 0.03; see Figure A6B), and PD-L1 TPS > 50% (HR 0.64, 95% CI 0.51–0.81, *p* = 0.0002; see Figure A6F), but not in patients with squamous mNSCLC (HR 0.87, 95% CI 0.74–1.01, *p* = 0.08; see Figure A5B) or PD-L1 TPS 1–49% (HR 0.97, 95% CI 0.82–1.15, *p* = 0.76; see Figure A6D).

### 3.4. Heterogeneity and Transitivity Assessments

Network meta-analysis is only valid if the interstudy heterogeneity is low (I^2^ < 50%) with acceptable transitivity. The majority of the meta-analyses performed had low levels of heterogeneity, and transitivity was largely acceptable as only selected RCTs were utilized. There were a few studies with I^2^ > 50%. The meta-analyses with substantial heterogeneity are described below.

There was substantial heterogeneity in the mOS (I^2^ = 67%, *p* = 0.02), mPFS (I^2^ = 92%, *p* = 0.00001), and ORR (I^2^ = 87%, *p* = 0.00001) meta-analysis for anti-PD-1 mAb monotherapy (see Figure A2, Figure A3 and Figure A4). EMPOWER Lung-1 and KEYNOTE 024 were identified as the outliers. Unlike the other studies, these two studies only included patients with PD-L1 TPS >50%. Pairwise meta-analyses were then performed for patients with PD-L1 TPS >50%, and this reduced the heterogeneity level to low-moderate (I^2^ = 36%, *p* = 0.18, Figure 3) for mOS, but remained substantial for mPFS (I^2^ = 78%) and ORR (I^2^ = 81%).

We detected a moderate and substantial level of heterogeneity in the mPFS meta-analyses for anti-PD-1 mAb + chemotherapy (I^2^ = 43%, *p* = 0.05; see Figure 4A) and anti-PD-L1 mAb + chemotherapy versus chemotherapy (I^2^ = 66%, *p* = 0.02; see Figure 4B), respectively. Sensitivity analysis revealed CheckMate 227 and GEMSTONE 302 contributed to the heterogeneity and were largely within the PD-L1 TPS < 1% population (I^2^ = 71%, *p* = 0.008, see Figure A7). Removing these trials significantly reduced the heterogeneity (anti-PD-1 mAb + chemotherapy: I^2^ = 24%, *p* = 0.2; anti-PD-L1 mAb + chemotherapy: I^2^ = 2%, *p* = 0.38).

### 3.5. Network Meta-Analyses for Survival and Tumor Response

Using the pairwise meta-analyses results from above, indirect comparisons were made between anti-PD-1 and anti-PD-L1 mAbs ± chemotherapy (Figure 5). To ensure transitivity and consistency, we only used meta-analyses with a low to moderate level of heterogeneity (i.e., I^2^ < 50%) for our NWM.

In the presence of chemotherapy, anti-PD-1 mAbs produced significantly better mOS compared with anti-PD-L1 mAbs (HR 0.81, 95% CI 0.71–0.91, *p* = 0.0006), but not as monotherapies (HR 0.91, 95% CI 0.74–1.13, *p* = 0.39). Subgroup NWM analyses were performed, and we demonstrated the superiority of anti-PD-1 mAb in the PD-L1 TPS 1–49% population (HR 0.63, 95% CI 0.50–0.79, *p* = 0.0001) and non-squamous histology (HR 0.81, 95% CI 0.67–0.97, *p* = 0.02 but not in PD-L1 TPS < 1% (HR 0.89, 95% CI 0.73–1.09, *p* = 0.27), PD-L1 TPS >50% HR 1.03, 95% CI 0.76–1.40, *p* = 0.84) and not significant for squamous histology (HR 0.79, 95% CI 0.63–1.01, *p* = 0.056; result not shown). We also demonstrated that anti-PD-1 mAb + chemotherapy had better mOS than anti-PD-1 mAb alone (HR 0.81, 95% CI 0.67–0.97, *p* = 0.02), but the mOS for anti-PD-L1 mAb + chemotherapy and anti-PD-L1 mAb alone were equivalent (HR 0.94, 95% CI 0.83–1.08, *p* = 0.39).

In the presence of chemotherapy, anti-PD-1 mAbs also had significantly better mPFS than anti-PD-L1 mAbs (even with GEMSTONE 302 included in the NWM, there was a significant difference between anti-PD-1 and anti-PD-L1 mAbs in mPFS (HR 0.84, 95% CI 0.71–1.00, *p* = 0.045) (HR 0.63, 95% CI 0.50–0.79, *p* = 0.0001) and ORR (OR 1.30, 95% CI 1.02–1.64, *p* = 0.03)). Consistent with mOS, mPFS was also significantly better in anti-PD-1 mAb + chemotherapy (HR 0.79, 95% CI 0.63–0.99, *p* = 0.04) in the PD-L1 TPS 1–49% population. Grade 3 and higher toxicities were not significantly different between anti-PD-1 mAbs and anti-PD-L1 mAbs with chemotherapy (OR 0.77, 95% CI 0.48–1.24, *p* = 0.29).

## 4. Discussion

This is the largest NWM comparing the efficacy of anti-PD-1 and anti-PD-L1 mAbs in mNSCLC. Our NWM has three main advantages over the previously published NWMs: We included a larger number of RCTs, giving more statistical power; we included 7 different anti-PD-1 mAbs and 4 different anti-PD-L1 mAbs, allowing comparison of anti-PD-1 to anti-PD-L1 mAbs as drug classes; we excluded studies with bevacizumab (IMpower 150 [9] and TASUKI 52 [47]), allowing a similar control arm for a more precise NWM.

In our meta-analyses, we confirmed the previous observation that anti-PD-1 with or without chemotherapy improves outcome and efficacy compared with chemotherapy alone for patients with mNSCLC without sensitizing driver mutations in the first-line setting. Interestingly, anti-PD-L1 with chemotherapy was equivalent in outcome and efficacy compared with chemotherapy in patients with PD-L1 TPS 1–49% and squamous histology (Table 2). This indicates that the tumor microenvironment and/or PD-L1 expression in these populations are heterogeneous and more prone to alteration, thus reducing anti-PD-L1 + chemotherapy’s efficacy. Numerically, the OS outcomes for anti-PD-1 or anti-PD-L1 mAb monotherapies were better in patients expressing high levels of PD-L1 in their tumors (TPS > 50%). In these patients, whether to add chemotherapy to the anti-PD-1 or anti-PD-L1 mAb remains debatable. A NWM published in 2021 has shown that in the PD-L1 TPS > 50% population, chemoimmunotherapy had a similar mOS compared with immunotherapy alone [48], suggesting that adding chemotherapy in this population may be unnecessary and preferable to minimizing toxicity. However, this study did not distinguish anti-PD-1 from anti-PD-L1 mAbs. Intriguingly, we showed that in order to achieve superior OS, chemotherapy is necessary in the setting of anti-PD-1 mAbs. Median OS was improved by 19% with anti-PD-1 mAb when combined with chemotherapy versus anti-PD-1 mAb monotherapy in patients with PD-L1 TPS > 50%. Of importance, this was not observed when anti-PD-L1 mAbs were used. Furthermore, in the presence of chemotherapy, anti-PD-1 mAbs were more effective than anti-PD-L1 mAbs. Compared with the anti-PD-L1 mAb + chemotherapy combination, anti-PD-1 mAb + chemotherapy improved mOS by 33%, mPFS by 22%, and response rate by 30%, with similar grade 3 or higher toxicities.

Together, these findings suggest chemotherapy may create a condition more favorable for anti-PD-1 mAbs as a plausible explanation. Chemotherapy has been recognized to favorably alter the immune TME in several ways: it increases immunogenic cell death [49], it improves dendritic cell maturation [50], it reduces the presence of immunosuppressive cells [51], it alters the expression of PD-L1 [52], and it increases the number of tumor-infiltrating lymphocytes [53]. However, these mechanisms should impact anti-PD-1 and anti-PD-L1 mAbs equally and do not explain the differences we have observed between the two mAbs in our NWM. A mechanistic difference between the two classes of mAbs will explain our observations.

Although both anti-PD-1 and anti-PD-L1 block the PD-1-PD-L1 interaction, anti-PD-1 and anti-PD-L1 mAbs are mechanistically unique. One of the key differences is that anti-PD-1 mAbs can block the interaction between PD-1 and both PD-L1 and PD-L2, while anti-PD-L1 mAbs cannot. PD-L2 is another ligand for PD-1, and its significance within the TME remains unclear. Several differences exist between PD-L1 and PD-L2. Firstly, PD-L1 is thought to be the main driver of T cell inhibition, as most tumors express higher levels of PD-L1 than PD-L2 [54,55]. This may explain why, when used as monotherapies, anti-PD-1 mAbs do not result in superior efficacy over anti-PD-L1 mAbs. Secondly, PD-L1 and PD-L2 expression do not correlate with one another. Poorly-differentiated tumors have been suggested to express more PD-L1 than PD-L2, while well-differentiated tumors express more PD-L2 than PD-L1 [56]. This differential expression of PD-L1 would be expected to impact the efficacy of anti-PD-L1 mAbs but not the efficacy of anti-PD-1 mAbs. Finally, PD-L2 binds to PD-1 with a higher affinity. If PD-L2 expression increases, it could outcompete PD-L1, making PD-L2 the predominant driver of immunosuppression over PD-L1. Chemotherapy is already known to upregulate PD-L2 expression on tumor cells [57] and is often more active in poorly differentiated tumors than well-differentiated tumors [58,59]. In the presence of chemotherapy, the TME might develop a higher PD-L2 to PD-L1 ratio. Out of the PD-L1 low (TPS < 1%), moderate (TPS 1–49%), and high (TPS > 50%) populations, only the PD-L1 moderate population is impacted by chemotherapy, suggesting this group of patients might have a more malleable TME for PD-L2 expression. Interestingly, a pooled analysis of two RCTs (PEMBRO-RT [60] and MDACC [61]) also showed that in the presence of radiotherapy, pembrolizumab is more effective, especially in the PD-L1 moderate population [62]. These observations warrant more investigations into the mechanisms that influence the tumor microenvironment of the PD-L1 TPS 1–49% population.

### 4.1. Implications

In our NWM, we have shown anti-PD-1 mAbs perform better than anti-PD-L1 mAbs in the presence of chemotherapy in patients with mNSCLC without sensitizing driver mutations in the first-line setting, especially for tumors expressing PD-L1 TPS 1–49%. In patients with mNSCLC who are suitable for chemotherapy, chemotherapy should be included in the regimen with anti-PD-1 mAb as it improves outcomes even in the PD-L1-high population. This has implications not just for mNSCLC but also for resectable NSCLC. In recent years, the appropriate neoadjuvant chemoimmunotherapy combination for resectable NSCLC has been investigated. Presented at the European Society of Medical Oncology (ESMO) in 2022, a phase II study showed that chemotherapy + nivolumab produced significantly better response rates than nivolumab alone, even in the PD-L1 high population (complete and major pathological response rates were 50% and 80% versus 18.2%, respectively) [63]. Consistent with our observation, chemotherapy was indeed required for a better response when combined with an anti-PD-1 mAb.

Furthermore, if PD-L2 indeed plays a role in the TME when chemotherapy is involved, then studies combining ICIs in conjunction with chemoradiotherapy may need to be performed using an anti-PD-1 mAb to achieve optimal disease-recurrence-free survival. Indeed, KEYNOTE 799 [64] already showed a numerically better ORR than the PACIFIC trial [65] (70.5% versus 28.4%, respectively). A RCT involving anti-PD-1 mAbs will be needed to confirm KEYNOTE 799’s findings.

In most trials, PD-L2 status is rarely reported, and it is difficult to determine if PD-L2 expression changes with chemotherapy administration. We recommend future clinical trials include PD-L2 analysis to give us a better understanding of the PD-1-PD-L2 interaction and confidence in the mAb being chosen. Furthermore, basic research is needed to better understand the complex interplay within the TME.

### 4.2. Limitations and Future Directions

There are several limitations to our study. Firstly, our indirect comparisons between anti-PD-1 and anti-PD-L1 mAbs do not replace a comparative prospective randomized study. Unfortunately, a RCT using all of the available anti-PD-1 and anti-PD-L1 mAbs would not be feasible or favorable to the industry. Thus, this NWM may offer the best evidence for the superiority of anti-PD-1 mAbs. Secondly, one of the important factors that could cause heterogeneity in our study was the inconsistent use of anti-PD-L1 antibodies used in IHC for PD-L1 TPS. Therefore, the difference in efficacy observed between the two mAbs by subgroup analyses using PD-L1 TPS needs to be interpreted with caution. Thirdly, we are still waiting for the OS results from RATIONALE 304 and 307. These results will add more power to our analyses once the data are more mature.

There are several questions left unanswered by our NWM: Is chemotherapy still necessary in the PD-L1 TPS > 90% population? According to 3-year update from correlative analysis presented at the American Society for Clinical Oncology (ASCO) in 2022, the ORR for pembrolizumab in PD-L1 TPS > 90% is 47.9% [66]. Interestingly, KEYNOTE 189 showed chemoimmunotherapy could achieve an ORR of 61.4% in the PD-L1 TPS > 50% population [8]. These results indicate that chemotherapy may still be beneficial regardless of PD-L1 status;Are anti-PD-1 mAbs better than anti-PD-L1 mAbs in mNSCLC with sensitizing driver mutations or in subsequent line therapies?Are anti-PD-1 mAbs superior to anti-PD-L1 in combination with bevacizumab, TKIs, or other ICIs?


## 5. Conclusions

In conclusion, anti-PD-1 mAbs appear to be superior to anti-PD-L1 mAbs in combination with chemotherapy for mNSCLC in the first-line setting. The TME is a complex entity that can be influenced by multiple factors, including chemotherapy. A head-to-head comparison between anti-PD-1 and anti-PD-L1mAbs in a clinical trial is needed to confirm our findings. Importantly, basic science research to further understand the mechanisms that influence the TME is needed to optimize informed trial design with biomarker selection and deliver reliable therapies in the future to improve survival in this insidious disease. 

## Figures and Tables

**Figure 1 biomedicines-11-01827-f001:**
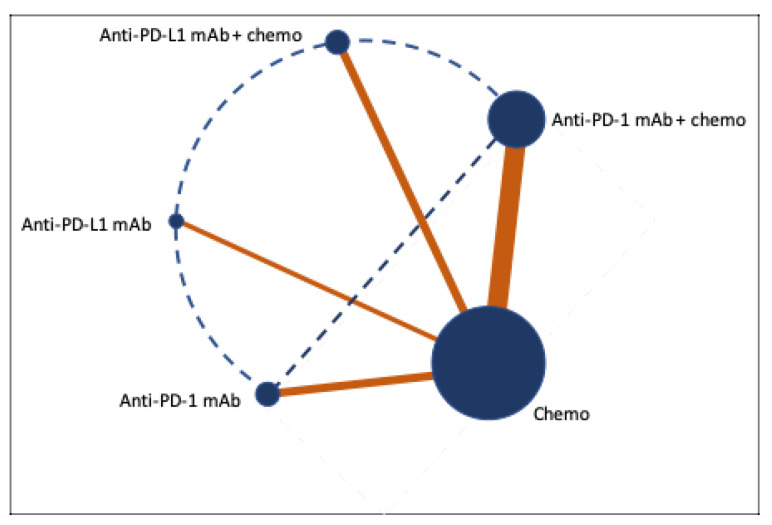
Network meta-analysis plot. A Bayesian framework was used to generate comparisons between the median overall survival, the median progression-free survival, and the objective response rate. The size of the blue circle and thickness of the orange lines are proportional to the number of studies included. Solid lines represent meta-analyses, and dotted lines represent network meta-analyses performed.

**Figure 2 biomedicines-11-01827-f002:**
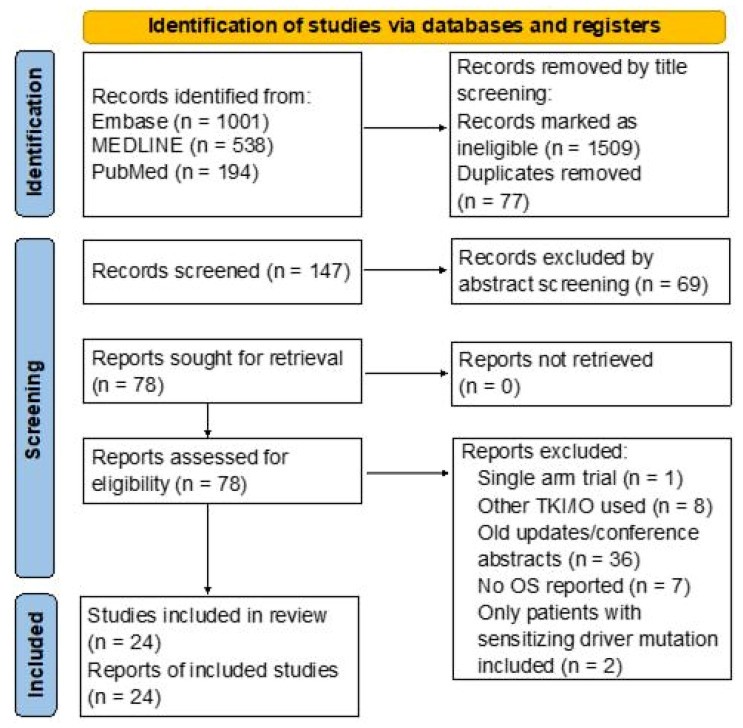
PRISMA flow diagram: Literature search and selection following PRISMA guidelines. TKI, tyrosine kinase inhibitor; IO, immunotherapy; OS, overall survival; EGFR, epithelial growth factor receptor; ALK, anaplastic lymphoma kinase; ROS-1, ROS proto-oncogene 1 receptor kinase.

**Figure 3 biomedicines-11-01827-f003:**
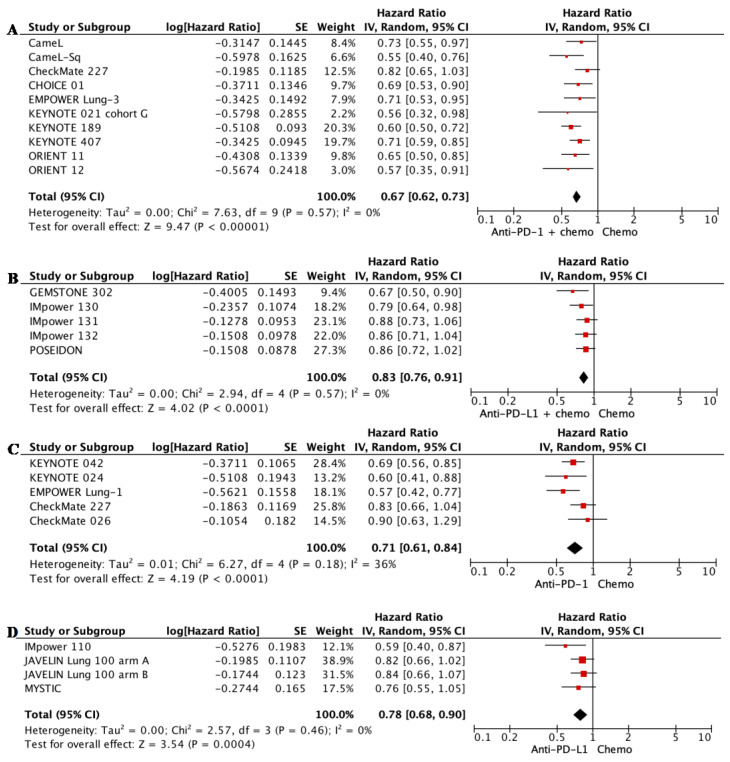
Forest plot for meta-analyses of median overall survival (mOS). (**A**) anti-PD-1 mAb + chemotherapy versus chemotherapy alone for all patients; (**B**) anti-PD-L1 mAb + chemotherapy versus chemotherapy alone for all patients; (**C**) anti-PD-1 mAb monotherapy versus chemotherapy alone for patients with PD-L1 TPS > 50%; (**D**) anti-PD-L1 mAb monotherapy versus chemotherapy alone for patients with PD-L1 TPS > 50%. Squares represent HRs of individual studies; rhombi represent HRs of meta-analyses.

**Figure 4 biomedicines-11-01827-f004:**
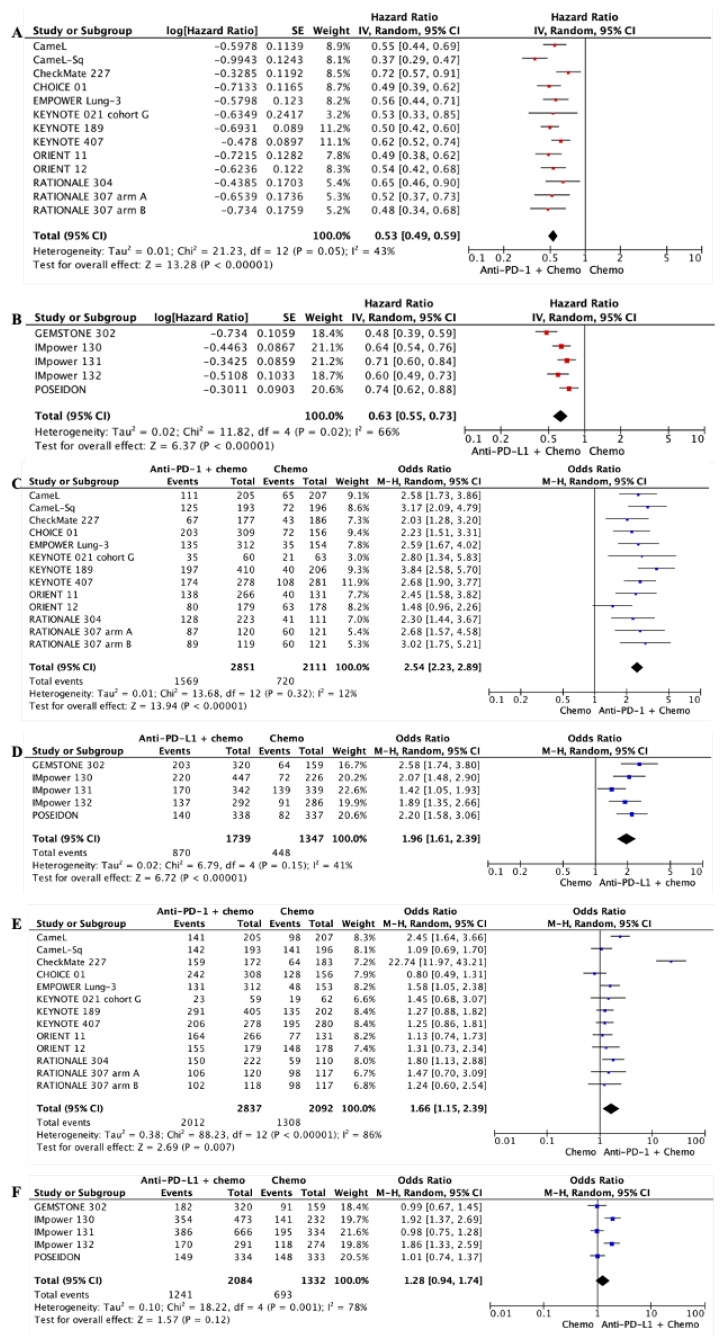
Forest plot for meta-analyses of median progression-free survival (mPFS), objective response rate (ORR), and grade 3 and higher toxicities for chemoimmunotherapies. (**A**) mPFS, anti-PD-1 mAb + chemotherapy versus chemotherapy alone; (**B**) mPFS, anti-PD-L1 mAb + chemotherapy versus chemotherapy alone; (**C**) ORR, anti-PD-1 mAb + chemotherapy versus chemotherapy alone; (**D**) ORR, anti-PD-L1 mAb + chemotherapy versus chemotherapy alone; grade 3 and higher toxicities for anti-PD-1 mAb + chemotherapy (**E**) and anti-PD-L1 mAb + chemotherapy (**F**) versus chemotherapy alone. Squares represent HRs or ORs of individual studies; rhombi represent meta-analyses.

**Figure 5 biomedicines-11-01827-f005:**
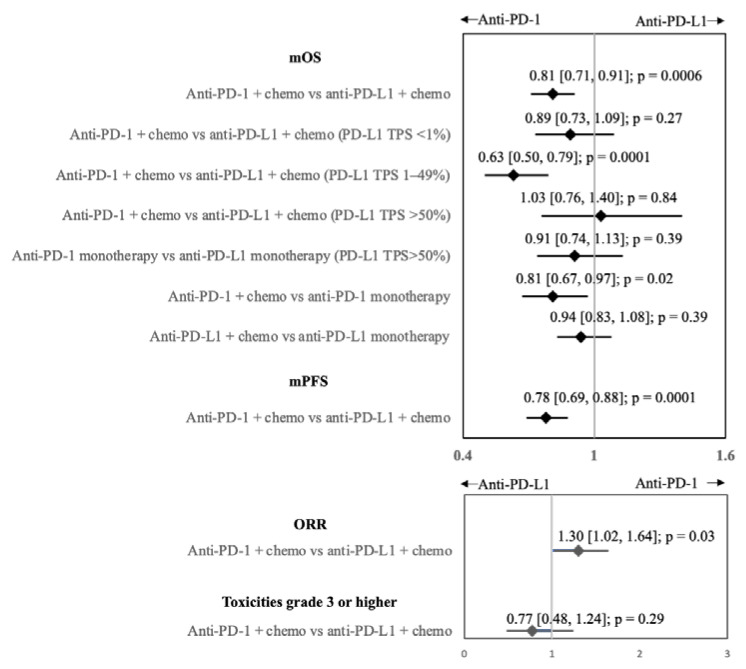
Forest plot for network meta-analyses of median overall survival (mOS), median progression-free survival (mPFS), objective response rate (ORR), and grade 3 or higher toxicities.

**Table 1 biomedicines-11-01827-t001:** Study characteristics of studies included in the network meta-analyses.

Study No.	Study Name	Number of Patients	Study Arms	Lung Cancer Histology Type	PD-L1 Status	Antibody Used to Determine PD-L1 Status
1	KEYNOTE 021 cohort G	ChemoIO: 60 Chemo: 63	Arm A: Pembrolizumab + Platinum doublet Arm B: Platinum doublet	Non-squamous	PD-L1 < 1%, 1–49%, >50%	22C3 pharmDX
2	KEYNOTE 189	ChemoIO: 410 Chemo: 206	Arm A: Pembrolizumab + Platinum doublet Arm B: Platinum doublet	Non-squamous	PD-L1 < 1%, 1–49%, >50%	22C3 pharmDX
3	KEYNOTE 407	ChemoIO: 278 Chemo: 281	Arm A: Pembrolizumab + Platinum doublet Arm B: Platinum doublet	Squamous	PD-L1 < 1%, 1–49%, >50%	22C3 pharmDX
4	CheckMate 227	ChemoIO: 177 IO alone: 396 Chemo: 397	Arm A: Nivolumab + ipilimumab + Platinum doublet Arm B (PD-L1 < 1%): nivolumab + Platinum doublet Arm C (PD-L1 > 1%): nivolumab alone Arm D: Platinum doublet	Non-small cell	PD-L1 < 1%, 1–49%, >50%	28-8 pharmDX
5	CHOICE 01	ChemoIO: 309 Chemo: 156	Arm A: Toripalimab + Platinum doublet Arm B: Platinum doublet	Non-small cell	PD-L1 < 1%, 1–49%, >50%	JS311 MEDx
6	CameL	ChemoIO: 205 Chemo: 207	Arm A: Camrelizumab + Platinum doublet Arm B: Platinum doublet	Non-squamous	PD-L1 < 1%, 1–49%, >50%	22C3 pharmDX
7	CameL-Sq	ChemoIO: 193 Chemo: 196	Arm A: Camrelizumab + Platinum doublet Arm B: Platinum doublet	Squamous	PD-L1 < 1%, 1–49%, >50%	E1L3N AmoyDx
8	EMPOWER-Lung 3	ChemoIO: 312 Chemo: 154	Arm A: Cemiplimab + Platinum doublet Arm B: Platinum doublet	Non-small cell	PD-L1 < 1%, 1–49%, >50%	SP263
9	ORIENT 11	ChemoIO: 266 Chemo: 131	Arm A: Sintilimab + Platinum doublet Arm B: Platinum doublet	Non-squamous	PD-L1 < 1%, 1–49%, >50%	22C3 pharmDX
10	ORIENT 12	ChemoIO: 179 Chemo: 178	Arm A: Sintilimab + Platinum doublet Arm B: Platinum doublet	Squamous	PD-L1 < 1%, 1–49%, >50%	22C3 pharmDX
11	RATIONALE 304	ChemoIO: 223 Chemo: 111	Arm A: Tislelizumab + Platinum doublet Arm B: Platinum doublet	Non-squamous	PD-L1 < 1%, 1–49%, >50%	SP263
12	RATIONALE 307	ChemoIO Arm A: 120 ChemoIO Arm B: 119 Chemo: 121	Arm A: Tislelizumab + Platinum + Paclitaxel Arm B: Tislelizumab + Platinum + Nab-paclitaxel Arm C: Platinum + Paclitaxel	Squamous	PD-L1 < 1%, 1–49%, >50%	SP263
13	IMpower 130	ChemoIO: 451 Chemo: 228	Arm A: Atezolizumab + Platinum doublet Arm B: Platinum doublet	Non-squamous	PD-L1 < 1%, 1–49%, >50% (TC) PD-L1 < 1%, 1–9%, >10% (IC)	SP142
14	IMpower 131	ChemoIO Arm A: 338 ChemoIO Arm B: 343 Chemo: 340	Arm A: Atezolizumab + Platinum + Paclitaxel Arm B: Atezolizumab + Platinum + Nab-paclitaxel Arm C: Platinum + Nab-paclitxel	Squamous	PD-L1 < 1%, 1–49%, >50% (TC) PD-L1 < 1%, 1–9%, >10% (IC)	SP142
15	IMpower 132	ChemoIO: 292 Chemo: 286	Arm A: Atezolizumab + Platinum doublet Arm B: Platinum doublet	Non-squamous	PD-L1 < 1%, 1–49%, >50% (TC) PD-L1 < 1%, 1–9%, >10% (IC)	SP142
16	GEMSTONE 302	ChemoIO: 320 Chemo: 159	Arm A: Sugemalimab + Platinum doublet Arm B: Platinum doublet	Non-small cell	PD-L1 < 1%, 1–49%, >50%	SP263
17	POSEIDON	ChemoIO: 338 Chemo: 337	Arm A: Durvalumab + Platinum doublet Arm B: Platinum doublet	Non-small cell	PD-L1 < 1%, 1–49%, >50%	SP263
18	Checkmate 026	ChemoIO: 271 Chemo: 270	Arm A: Nivolumab Arm B: Platinum doublet	Non-small cell	PD-L1 > 5%, >50%	28-8 pharmDX
19	Keynote 024	ChemoIO: 154 Chemo: 151	Arm A: Pembrolizumab Arm B: Platinum doublet	Non-small cell	PD-L1 > 50%	22C3 pharmDX
20	Keynote 042	ChemoIO: 637 Chemo: 636	Arm A: Pembrolizumab Arm B: Platinum doublet	Non-small cell	PD-L1 1–49%, >50%	22C3 pharmDX
21	EMPOWER-Lung 1	ChemoIO: 356 Chemo: 354	Arm A: Cemiplimab Arm B: Platinum doublet	Non-small cell	PD-L1 >50%	22C3 pharmDX
22	IMpower 110	ChemoIO: 227 Chemo: 227	Arm A: Atezolizumab Arm B: Platinum doublet	Non-small cell	PD-L1 > 1%, >5%, >50%	SP142
23	JAVELIN Lung 100	ChemoIO Arm A: 366 ChemoIO Arm B: 322 Chemo: 526	Arm A: Avelumab Q2W Arm B: Avelumab QW Arm C: Platinum doublet	Non-small cell	PD-L1 > 1%, >50%, >80%	73-10 pharmDx
24	MYSTIC	ChemoIO Arm A: 374 ChemoIO Arm B: 372 Chemo: 372	Arm A: Durvalumab Arm B: Durvalumab + Tremelimumab Arm C: Platinum doublet	Non-small cell	PD-L1 > 25%	SP263

**Table 2 biomedicines-11-01827-t002:** Summary of overall survival results from meta-analyses.

Treatment		Hazard Ratio (95% CI)
Anti-PD-1 mAbs + chemotherapy > chemotherapy	All patients	0.67 (0.62–0.73)
	PD-L1 TPS < 1%	0.76 (0.66–0.87)
	PD-L1 TPS 1–49%	0.61 (0.52–0.71)
	PD-L1 TPS > 50%	0.66 (0.54–0.81)
	Squamous	0.69 (0.58–0.83)
	Non-squamous	0.67 (0.58–0.76)
Anti-PD-L1 + chemotherapy > chemotherapy	All patients	0.83 (0.76–0.91)
	PD-L1 TPS < 1%	0.85 (0.74–0.99)
	PD-L1 TPS > 50%	0.64 (0.51–0.81)
	Non-Squamous	0.83 (0.73–0.93)
Anti-PD-L1 + chemotherapy = chemotherapy	PD-L1 TPS 1–49%	0.97 (0.82–1.15)
	Squamous	0.87 (0.74–1.01)
Anti-PD-1 > chemotherapy	PD-L1 TPS > 1%	0.83 (0.70–0.97)
	PD-L1 TPS > 50%	0.71 (0.61–0.84)
Anti-PD-L1 > chemotherapy	PD-L1 TPS > 1%	0.88 (0.79–0.99)
	PD-L1 TPS > 50%	0.78 (0.68–0.90)

## Data Availability

Data used in this network meta-analysis are collected from published clinical trials.

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
