# Peer review of "Anti-PD-1 Monoclonal Antibodies (mAbs) Are Superior to Anti-PD-L1 mAbs When Combined with Chemotherapy in First-Line Treatment for Metastatic Non-Small Cell Lung Cancer (mNSCLC): A Network Meta-Analysis"

_biomedicines, 2023, doi:10.3390/biomedicines11071827_

Round 1
Reviewer 1 Report
Authors demonstrated that Anti-PD-1 antibodies are superior to anti-PD-L1 antibodies in first-line treatment for metastatic non-small cell lung cancer, using network meta-analysis.
The validity of the data contained in this manuscript should be assessed by a statistician, and as I am a clinician, I will only assess the Discussion part.
If interaction with PD-L2 is the reason behind PD-1 antibodies being more effective than PD-L1 antibodies, then PD-1 antibodies are also likely to have more side effects than PD-L1 antibodies. It is not fair to discuss only efficacy without discussing side effects.
Besides, three unresolved problems are presented as limitations in the Discussion part, but it is meaningless to state this as the study was not designed to solve these problems in the first place.
Author Response
Dear Reviewer,
Thank you for reviewing our manuscript. Please see below responses to your suggestions.
- If interaction with PD-L2 is the reason behind PD-1 antibodies being more effective than PD-L1 antibodies, then PD-1 antibodies are also likely to have more side effects than PD-L1 antibodies. It is not fair to discuss only efficacy without discussing side effects.
We have now added analyses for grade 3 or higher toxicities
- Besides, three unresolved problems are presented as limitations in the Discussion part, but it is meaningless to state this as the study was not designed to solve these problems in the first place.
We have changed this to future indications rather than limitations to our study.
Kind regards,
Joe Wei
Reviewer 2 Report
A huge, excellent, job has been done. Very important results were obtained. Congratulations to the authors.
Comments:
• I suggest the most important results (overall survival) summarize in a table, e.g.:
Chemotherapy + anti-PD-1 mAb > Chemotherapy (all histological types);
anti-PD-1 mAb = Chemotherapy;
anti-PD-L1 mAbs = Chemotherapy.
(or in another simple form).
• I suggest in the Discussion part these most important results summarize and discuss sequentially.
• Discussion part statement "In our meta-analyses, we confirmed the previous observation that anti-PD-1 or anti-291 PD-L1 mAbs with or without chemotherapy improve outcome and efficacy compared with chemotherapy alone for patients with mNSCLC in the first-line setting." (Lines 291-293) is only partly based on the results of the work, and partly contradicts the results.
• The same for 4.1. Implications section‘s first sentence "In our NWM, we have shown anti-PD-1 mAbs perform better than anti-PD-L1 mAbs 346 in the presence of chemotherapy." (Lines 346-347). The statement is not accurate. It should be indicated in which clinical cases are "better".
• For the abstract. Because a lot of data was obtained and even all the most important ones cannot be contained in an abstract, I would suggest presenting the information in a more summarized way.
Author Response
Dear reviewer,
Thank you for your positive feedback. Please see below addressing your suggestions.
• I suggest the most important results (overall survival) summarize in a table, e.g.:
Chemotherapy + anti-PD-1 mAb > Chemotherapy (all histological types);
anti-PD-1 mAb = Chemotherapy;
anti-PD-L1 mAbs = Chemotherapy.
(or in another simple form).
• I suggest in the Discussion part these most important results summarize and discuss sequentially.
Now the results are summarized in Table 2 and added discussion.
• Discussion part statement "In our meta-analyses, we confirmed the previous observation that anti-PD-1 or anti-291 PD-L1 mAbs with or without chemotherapy improve outcome and efficacy compared with chemotherapy alone for patients with mNSCLC in the first-line setting." (Lines 291-293) is only partly based on the results of the work, and partly contradicts the results.
Added “In our meta-analyses, we confirmed the previous observation that anti-PD-1 with or without chemotherapy improve outcome and efficacy compared with chemotherapy alone for patients with mNSCLC without sensitizing driver mutations in the first-line setting. Interestingly, anti-PD-L1 with chemotherapy was equivalent in outcome and efficacy compared with chemotherapy in patients with PD-L1 TPS 1–49% and squamous histology (Table 2).”
• The same for 4.1. Implications section‘s first sentence "In our NWM, we have shown anti-PD-1 mAbs perform better than anti-PD-L1 mAbs 346 in the presence of chemotherapy." (Lines 346-347). The statement is not accurate. It should be indicated in which clinical cases are "better".
This now reads “In our NWM, we have shown anti-PD-1 mAbs perform better than anti-PD-L1 mAbs in the presence of chemotherapy in patients with mNSCLC without sensitizing driver mutations in the first-line setting, especially for tumors expressing PD-L1 TPS 1–49%.”
• For the abstract. Because a lot of data was obtained and even all the most important ones cannot be contained in an abstract, I would suggest presenting the information in a more summarized way.
Abstract now written in a more summarized way with important data included now.
Kind regards,
Joe Wei
Reviewer 3 Report
In their manuscript, Wei and collaborators performed a network meta-analysis to compare the performance of PD-1 vs. PDL-1 antibodies in combination with chemotherapy as first line treatments in mNSCLC. To do this, they compared OS, PFS and ORR in 24RCTs. While the authors found that the antibodies were equivalent as single agents, they found that the combination of PD-1 mAb + chemotherapy showed superior mOS, mPFS, and ORR, compared to PD-L1 mAb + chemotherapy.
While the manuscript does not answer every question regarding the differences in response to the PD-1 vs PD-L1 mAbs when combined to chemotherapy, this study is an in-depth comparison of the efficacy of the two treatments based on available clinical data that is unique. The results are clearly presented and the text is very easy to follow. I do not have any recommendation aside from the correction of minor typos.
Nothing major aside from minor typos.
Author Response
Dear reviewer,
Thank you very much for your positive response.
Kind regards,
Joe Wei